# On-Treatment Decrease of Serum Interleukin-6 as a Predictor of Clinical Response to Biologic Therapy in Patients with Inflammatory Bowel Diseases

**DOI:** 10.3390/jcm9030800

**Published:** 2020-03-15

**Authors:** Gian Paolo Caviglia, Chiara Rosso, Francesco Stalla, Martina Rizzo, Alessandro Massano, Maria Lorena Abate, Antonella Olivero, Angelo Armandi, Ester Vanni, Ramy Younes, Sharmila Fagoonee, Rinaldo Pellicano, Marco Astegiano, Giorgio Maria Saracco, Elisabetta Bugianesi, Davide Giuseppe Ribaldone

**Affiliations:** 1Department of Medical Sciences, University of Turin, 1016 Turin, Italy; chiara.rosso@unito.it (C.R.); marialorena.abate@unito.it (M.L.A.); antonella.olivero@unito.it (A.O.); ramy.younes@unito.it (R.Y.); giorgiomaria.saracco@unito.it (G.M.S.); elisabetta.bugianesi@unito.it (E.B.); 2Unit of Gastroenterology, Città della Salute e della Scienza di Torino-Molinette Hospital, 10126 Turin, Italy; rossoalfa92@hotmail.it (F.S.); rizzomartina.92@gmail.com (M.R.); ale95.massano@gmail.com (A.M.); armandiangelo91@gmail.com (A.A.); evanni@cittadellasalute.to.it (E.V.); rinaldo_pellican@hotmail.com (R.P.); marcoastegiano58@gmail.com (M.A.); 3Institute of Biostructure and Bioimaging, CNR c/o Molecular Biotechnology Centre, 10126 Turin, Italy; sharmila.fagoonee@unito.it

**Keywords:** adalimumab, Crohn’s disease, cytokines, IL-6, sCD163, ulcerative colitis, ustekinumab, vedolizumab, zonulin

## Abstract

In patients with inflammatory bowel diseases (IBD) undergoing biologic therapy, biomarkers of treatment response are still scarce. This study aimed to evaluate whether serum zonulin, a biomarker of intestinal permeability; soluble CD163 (sCD163), a macrophage activation marker; and a panel of serum cytokines could predict the response to biologic treatment in patients with IBD. For this purpose, we prospectively enrolled 101 patients with IBD and 19 patients with irritable bowel syndrome (IBS) as a control group; 60 out of 101 patients underwent treatment with biologics. Zonulin, sCD163, and cytokines were measured at the baseline in all patients and after 10 weeks of treatment in the 60 patients who underwent biologic therapy. We observed that zonulin levels were higher in IBD patients with active disease compared to those in remission (*p* = 0.035), and that sCD163 values were higher in patients with IBD compared to those with IBS (*p* = 0.042), but no association with therapy response was observed for either biomarker. Conversely, interleukin (IL)-6, IL-8, IL-10, and tumor necrosis factor-alpha showed a significant reduction from baseline to week 10 of treatment, particularly in responder patients. By multivariate logistic regression analysis corrected for disease (Crohn’s disease or ulcerative colitis), type of biologic drug (Infliximab, Adalimumab, Vedolizumab, or Ustekinumab) and disease activity, the reduction in IL-6 values was associated with a clinical response at 12 months of biological therapy (odds ratio (OR) = 4.75, 95% confidence interval (CI) 1.25–18.02, *p* = 0.022). In conclusion, the measurement of serum IL-6 in biologics-treated IBD patients may allow for the prediction of response to treatment at 12 months of therapy and thus may help with tailoring personalized treatment strategies.

## 1. Introduction

Inflammatory bowel diseases (IBD) are chronic intestinal disorders consisting of two disease entities, Crohn’s disease (CD) and ulcerative colitis (UC), both characterized by an immune-mediated pathogenesis and a clinical relapsing course [1,2]. The exact etiology of IBD is unknown, but it is supposed that impaired intestinal permeability, together with genetic, microbial, and environmental factors, contributes to the onset of the disease [3,4,5]. The increased passage of non-self-antigens through the intestinal barrier may lead to the abrogation of immune tolerance in genetically susceptible individuals [6]; the activation of the innate immune system, followed by the adaptive immune response, are responsible for the initiation and maintenance of chronic inflammation and, thus, disease progression [7].

Zonulin is a modulator intercellular tight junction involved in the regulation of intestinal permeability. It has been shown that serum zonulin levels are increased in patients with IBD compared to healthy controls [8] (while no studies are present in the literature comparing levels in patients with IBD and levels in patients with gastrointestinal symptoms without clear inflammation, like patients affected by irritable bowel symptoms (IBS)) and are associated with higher levels of pro-inflammatory cytokines such as tumor necrosis factor-alpha (TNFα) and interleukin (IL)-6 [9], which play a key role in promoting mucosal inflammation [10]. Activated macrophages contribute to the immunological response as well. Increased levels of circulating soluble(s)-CD163, a macrophage activation marker, was associated with active CD and UC showing higher values in IBD patients compared to healthy subjects; in addition, soluble CD163 (sCD163) values showed a decrease over the course of treatment, reaching levels similar to those observed in the healthy population [11].

In the last few decades, a more comprehensive understanding of the cytokine pathways involved in the pathogenesis of IBD has allowed for the development of new treatment strategies that have permitted us to reduce the use of corticosteroids [12]. The chimeric antibody against TNFα, namely infliximab (IFX), represents the herald of a class of biologic drugs that target specific cytokines in the inflammatory cascade. Subsequently, several new biologic agents were introduced into clinical practice, including the less immunogenic humanized antibody against TNFα (i.e., adalimumab (ADA)), anti-integrin agents (i.e., vedolizumab (VDZ)), and anti-interleukin agents (i.e., ustekinumab (UTK)) [13]. These therapies proved effective at inducing disease remission, improving patient outcomes, and preventing progression to irreversible bowel damage [14]. However, only certain subgroups of patients with IBD receiving biologics appear to have beneficial clinical effects. Indeed, some patients do not respond to induction therapy or lose response during maintenance therapy after achieving an adequate induction response [15]. Therefore, it is crucial to identify biomarkers that can predict and monitor therapeutic success in order to tailor individualized treatment strategies. Hence, the primary aim of our study was to assess whether serum zonulin, sCD163, and a panel of serum cytokines can predict the response to biologic treatment in patients with IBD. The secondary aim of the study was to compare biomarker levels between patients with IBD and patients with functional intestinal disorders.

## 2. Materials and Methods

### 2.1. Patients

Patients with a definite diagnosis of CD or UC according to the European Crohn’s and Colitis Organisationcriteria [16,17] were prospectively enrolled between January 2018 and January 2019 at the outpatient clinic of the Unit of Gastroenterology of Città della Salute e della Scienza di Torino (Molinette Hospital), Turin, Italy. A control group of patients with diarrhea and IBS was included in the study for a comparative cross-sectional analysis. Such patients were chosen as a control group since they share with IBD patients overlapping clinical conditions in spite of the distinct pathophysiological and prognostic features.

A full medical history was obtained from all patients enrolled, and data regarding gender, age, smoking habit, type and duration of disease, disease activity, and previous therapy were recorded. According to Montreal’s classification [18], CD was classified as follows: L1, disease confined to distal ileum; L2, disease confined to the colon; L3, ileocolonic location; L4, upper gastrointestinal tract (as a modifier for L1, L2, and L3). UC was classified according to the segment involved: E1, ulcerative proctitis; E2, left-sided colitis; E3, pancolitis. Disease activity was calculated for CD using the Harvey‒Bradshaw index and for UC with the partial Mayo score [19,20]. All patients underwent a physical examination, infectivologic screening (antibodies to hepatitis C virus, hepatitis B surface antigen, antibodies to hepatitis B core antigen, antibodies to hepatitis B surface antigen, quantiferon assay for tuberculosis screening, antibodies to human immunodeficiency virus, human papilloma virus test, antibodies to varicella-zoster virus, antibodies to Epstein-Barr virus), first-step hematology, and biochemistry tests, including fecal calprotectin (FC), erythrocyte sedimentation rate (ESR), and C-reactive protein (CRP). Pregnancy, age < 18 years, prescription of the biological drug only because of an extraintestinal manifestation, and absence of signed written informed consent were the exclusion criteria. All patients included underwent venous blood sampling; serum was collected in polypropylene 2-mL tubes labeled with the study participant’s identification code and stored at −80 °C until analysis.

Patients affected by IBD with indication to biologic treatment (moderate to severe disease activity or steroid-dependent disease, with previous failure or intolerance to thiopurines [16,17]) were included in the longitudinal analysis and serum samples were also collected at 10 weeks of therapy. Clinical response was assessed at 12 months of biologic therapy.

Study procedures were compliant with the principles of the Declaration of Helsinki. All patients gave their written informed consent and the study was approved by the Ethics Committee of the Città della Salute e della Scienza—University Hospital of Turin (approval code 0056924). 

### 2.2. Measurement of Serum Zonulin, sCD163, and Cytokines

Serum zonulin was assessed by competitive enzyme-linked immunosorbent assay (ELISA) (IDK^®^ Zonulin ELISA Kit, Immunodiagnostik AG, Bensheim, Germany) according to the manufacturer’s instructions. Concentrations were calculated using a four-parameter algorithm and the results were given in ng/mL, as previously reported [8]. Serum sCD163 was measured by the ELISA method (Quantikine^®^ ELISA Human CD163, R&D Systems, Minneapolis, MN, USA) and concentrations were calculated using a linear log/log curve fit. Results are reported in ng/mL. The cytokine panel, including IL-1β, IL-4, IL-6, IL-8, IL-10, IL-12(p70), IL-17, IL-23, IL-33, interferon-gamma (IFNγ), and TNFα, was measured in serum samples by Bio-Plex^®^ Multiplex Immunoassay (Bio-rad Laboratories, Pleasanton, CA, USA) on a Luminex^®^ 200 system (Luminex Corporation, Austin, TX, USA). Individual standard curves were generated for each cytokine; the results are given in pg/mL. Personnel performing laboratory investigations were blind to all the characteristics of the patients included in the study.

### 2.3. Outcomes

The primary outcome was the prediction of clinical response at 12 months of biologic therapy by the measurement of serum zonulin, sCD163, and the selected cytokines at baseline and at 10 weeks of treatment. The secondary outcome was the inference of clinical and biochemical characteristics of the included patients by the use of zonulin, sCD163, and the panel of cytokines.

According to the literature [21], clinical response to biologic therapy was defined as a decrease in the Harvey‒Bradshaw index (HBI) greater than or equal to 3 (or HBI ≤ 4 at month 12) or in the partial Mayo (pMAYO) score greater than or equal to 2 (or pMAYO ≤ 1 at month 12), in the absence of corticosteroid therapy. Patients who discontinued biologic treatment, or those lost to follow-up, were considered as cases of treatment failure (intention to treat analysis).

### 2.4. Statistical Analysis

Continuous variables were reported as median (range or 95% confidence interval (CI)) according to the data distribution. Normality was checked by the D’Agostino‒Pearson test. Categorical variables were reported as number and percentage. Comparison of continuous variables between independent groups was performed by the Mann‒Whitney test; comparison between paired measurements was performed by the Wilcoxon test. Correlation between variables was performed by the nonparametric Spearman correlation test. Regarding the dichotomous categorical variable, a Fisher’s exact test or McNemar test was performed for unpaired or paired analysis, respectively. The association between variables was assessed by logistic regression analysis; the strength of association was reported as the odds ratio (OR) and 95% CI.

All statistical analyses were performed using MedCalc^®^ v.18.9.1 (MedCalc Software Ltd., Ostend, Belgium), and a *p* value ≤ 0.05 was considered statistically significant.

## 3. Results

### 3.1. Cross-Sectional Analysis

A total of 120 patients (IBD: *n* = 101; IBS: *n* = 19) were included in the study. The demographic, clinical, and biochemical characteristics of the study population are reported in Table 1.

Patients with IBD were slightly older than those with IBS (*p* = 0.067), and the former showed a higher prevalence of males (61.3%) compared to the latter (31.6%) (*p* = 0.023). Among patients with IBD, the majority had a diagnosis of CD (71.3%); the median HBI was 6 (95% CI 5–6) in patients with CD, while the median pMAYO score was 4 (95% CI 3–5) in those with UC. The values of zonulin, sCD163, and cytokines according to the diagnosis of IBD or IBS are reported in Table 2.

As IL-1β, IL-4, IL-12(p70), IL-17, IL-23, and IFNγ were not quantifiable in almost all patients, no further analysis was performed on these cytokines. We observed that sCD163 and IL-8 levels were significantly increased in IBD patients compared to those with IBS, while among patients with IBD, serum zonulin levels were significantly different between patients in clinical remission and those with active disease (43.6 (95% CI 20.4–46.4) ng/mL vs. 47.4 (95% CI 43.6–48.9) ng/mL, *p* = 0.035) (Figure 1).

In patients with IBD, IL-6 values were correlated with years of disease duration (*r_s_* = −0.219 (−0.402 – −0.019), *p* = 0.032), CRP values (*r_s_* = 0.301 (0.091–0.485), *p* = 0.006), and FC concentration (*r_s_* = 0.331 (0.104–0.525), *p* = 0.005); IL-8 values resulted correlated only with FC (*r_s_* = 0.293 (0.062–0.494), *p* = 0.014) (Figure 2).

No other significant correlations were observed between zonulin, sCD163, and cytokine levels with demographic, clinical, and biochemical characteristics of the study population. Moreover, no differences were observed in zonulin, sCD163, and cytokine levels between patients with CD and those with UC (Appendix A). In the whole cohort of patients, we observed a strong positive correlation between IL-10 and IL-33 values (*r_s_* = 0.633 (95% CI 0.552–0.760), *p* < 0.001), IL-8 and TNFα (*r_s_* = 0.537 (95% CI 0.409–0.670), *p* < 0.001), IL-33 and TNFα (*r_s_* = 0.508 (95% CI 0.375–0.646), *p* < 0.001) and between TNFα and zonulin (*r_s_* = 0.316 (95% CI 0.150–0.483), *p* < 0.001). Conversely, IL-6 was negatively correlated with IL-10 (*r_s_* = −0.429 (95% CI −0.589 to −0.292), *p* < 0.001) and IL-33 (*r_s_* = −0.308 (95% CI −0.509 to −0.184), *p* < 0.001) (Figure 3).

### 3.2. Longitudinal Analysis

Among the whole study cohort, 60 patients with moderate to severe disease activity or steroid-dependent disease with previous failure or intolerance to thiopurines (median age 49 (18–80) years, males: *n* = 38, 63.4%) underwent biologic therapy. The majority of treated patients had a diagnosis of CD (*n* = 43, 71.7%). The median HBI was 6 (95% CI 5–7) for patients with CD, while the median pMAYO score was 5 (95% CI 3–7) for patients with UC. Among patients with CD, 31 underwent treatment with ADA, 10 with VDZ, and two with UTK; among patients with UC, four underwent treatment with IFX, two with ADA, and 11 with VDZ. Anti-TNF therapy was the first-line choice for all patients, except for the patients with contraindications to these drugs; VDZ or UTK were the second (or third) choice, except for patients with contraindications to anti-TNF therapy. Upon treatment initiation, 55 patients (91.7%) received mesalazine in combination, 25 patients (41.7%) took systemic corticosteroids, and none took immunosuppressants.

Clinical response at 12 months was observed in 32 patients (53.4%). In detail, 19 out of 33 patients (57.6%) responded to ADA, one out of four (25.0%) responded to IFX, one out of two (50%) responded to UTK, and 11 out of 20 (55.0%) responded to VDZ. Among the 28 non-responder patients, nine switched to other biologics, seven started corticosteroids for inducing or maintaining a clinical response, four underwent surgical resection, and two were lost to follow-up. The remaining six patients failed to achieve a reduction in HBI or pMAYO score.

In the overall cohort of patients that underwent biologic therapy, we observed a significant reduction from baseline to week 10 of treatment for IL-6, IL-8, IL-10, and TNFα (Table 3).

According to the treatment response, IL-6, IL-8, and TNFα decreased significantly only in responder patients, while IL-10 declined significantly in both responders and non-responders (Figure 4). At baseline, no significant differences were observed for zonulin, sCD163, and cytokine levels between responders and non-responders (Appendix A).

Concerning biomarkers of inflammation, CRP values decreased significantly from baseline to week 10 of treatment (CRP: 6.6 (95% CI 5.0–8.9) mg/L vs. 3.3 (95% CI 2.2–5.1) mg/L, *p* < 0.001). According to the treatment response, CRP values decreased significantly in responder patients (5.8 (95 % CI 4.0–25.2) mg/L vs. 3.2 (95% CI 1.9–5.3) mg/L, *p* = 0.003) while no significant changes were observed in non-responders (8.0 (95% CI 4.9–9.7) mg/L vs. 4.0 (95% CI 1.0–9.3) mg/L, *p* = 0.104) (Figure 5A).

Overall, FC values decreased significantly from baseline to week 10 of treatment (534 (95% CI 271–996) µg/g vs. 141 (95% CI 99–356) µg/g, *p* = 0.003). According to the treatment response, FC values decreased significantly in responder patients (600 (95 % CI 281–1212) µg/g vs. 100 (95% CI 80–359) µg/g, *p* = 0.003), while no significant decline was observed in non-responders (293 (95% CI 103–1845) µg/g vs. 176 (95% CI 40–858) µg/g, *p* = 0.305) (Figure 5B). Of note, no differences in CRP and FC levels were observed at baseline (*p* = 0.735 and *p* = 0.591, respectively) and at week 10 of treatment (*p* = 0.984 and *p* = 0.851, respectively) between responders and non-responders.

The reduction of CRP and FC values from baseline to week 10 of treatment was not significantly associated with a clinical response at 12 months of biologic therapy (CRP: OR = 0.71, 95% CI 0.19–2.69, *p* = 0.619; FC: OR = 4.44, 95% CI 0.68–28,86, *p* = 0.118). Conversely, a reduction in IL-6 values from baseline to week 10 of treatment was able to predict clinical response (OR = 3.89, 95% CI 1.11–13.68, *p* = 0.034). No association with clinical response was observed for the reduction of IL-8 (OR = 0.93, 95% CI 0.30–2.85, *p* = 0.895), IL-10 (OR = 1.55, 95% CI 0.51–4.65, *p* = 0.439), or TNFα (OR = 1.06, 95% CI 0.33–3.37, *p* = 0.927).

In a multivariate logistic regression analysis corrected for disease (CD or UC), type of biologic drug (ADA, IFX, VDZ, or UTK), and disease activity, the reduction of IL-6 values from baseline to 10 weeks of treatment remained a significant independent predictor of clinical response at 12 months of biological therapy (OR = 4.75, 95% CI 1.25–18.02, *p* = 0.022).

## 4. Discussion

Various of the findings of the present study are noteworthy. Firstly, serum zonulin, as biomarker of intestinal permeability, was elevated in patients with IBD and in those with IBS. In the former group, higher zonulin values were associated with disease activity, but no variation was observed in the course of biologic treatment in either responder or non-responder patients. Secondly, sCD163, a macrophage activation marker, was significantly higher in patients with IBD compared to those with IBS; however, no further association was observed with clinical features or response to biologic treatment. Finally, among the investigated cytokines, we observed that IL-6 was significantly correlated with standard biomarkers of inflammation (i.e., CRP and FC) and with disease duration, and varied significantly during biologic treatment. Remarkably, the decrease in IL-6 from baseline to week 10 of treatment was an independent predictor of clinical response at 12 months of biologic therapy.

Studies from animal models have highlighted that impaired intestinal permeability represents an early event preceding the onset of overt colitis [22]. In a previous study, we observed that patients with IBD had higher serum zonulin levels than healthy subjects [8]. Here, we observed similar values in an independent cohort of IBD patients. Interestingly, serum zonulin levels were not significantly different between patients with IBD and those with IBS, suggesting impaired intestinal permeability in patients with functional intestinal diseases. Accordingly, it has been shown that serum zonulin is upregulated in IBS [23], particularly in patients with diarrhea-predominant type [24]. Taken together, these observations are consistent with the hypothesis that increased intestinal permeability may be a necessary but not sufficient condition for the development of several intestinal diseases. As a matter of fact, we did not observe a variation in serum zonulin during the first 10 weeks of biologic treatment, even in responder patients; further studies are warranted to assess the possibility of restoring intestinal barrier functionality in patients with IBD in long-term remission.

Activated intestinal macrophages play a pivotal role in promoting the immunological response observed in IBD by producing pro-inflammatory cytokines such as IL-1β, IL-23, and TNFα [25,26], and by expressing a large number of T cell co-stimulating molecules [27]. In agreement with our findings, a previous study reported higher circulating levels of sCD163 in patients with IBD compared to healthy controls [11]. Another study showed that CD163 expression in colonic tissue samples was more pronounced in patients with IBD in comparison to non-IBD controls; furthermore, within the same IBD patient, CD163 was upregulated only in the inflamed tissue samples, compared to uninvolved mucosal samples [28]. Finally, we observed that biologic treatment did not affect circulating levels of sCD163 at 10 weeks of therapy. Consistently, Franzè et al. pointed out that in patients with IBD treated with mesalazine or corticosteroids or immunomodulators, CD163 RNA expression was not influenced by the ongoing treatments [28]. Conversely, Dige et al. found that anti-TNFα treatment already induced a rapid decrease in sCD163 levels one day after treatment initiation followed by a further decrease at week 1 and week 6, while in patients treated with prednisolone, sCD163 decline failed to reach statistical significance; however, no association was observed with clinical response [11]. It is likely that the differences among the clinical features of the patients included in the studies, the heterogeneity of the therapeutic approaches, the different study designs, and the assessment of CD163 mucosal expression rather than the circulating levels may explain the partially discordant results.

Among the investigated cytokines, notable results have been obtained for IL-6. The possibility of predicting clinical response to biological therapy would be an extremely valuable tool [29]. Moreover, considering that several anti-cytokine compounds are now available for the treatment of patients with IBD [30], the identification of a biomarker that could broadly predict therapy response irrespective of the type of biologic is of crucial relevance. Based on our results, patients with IBD showing a reduction in IL-6 from baseline to 10 weeks of biologic therapy have a 4.7-fold higher probability of achieving a clinical response at 12 months of biologic therapy compared to patients with no IL-6 reduction. Consistently, a recent study showed that serum patterns of IL-6 together with IL-8 at baseline and over the first six weeks of treatment with VDZ had 83% sensitivity and 87% specificity to predict mucosal healing, and 82% sensitivity and 90% specificity to predict clinical remission in UC patients [31]. In patients with UC undergoing IFX therapy, it has been shown that baseline serum IL-6 levels were significantly lower in responders than in non-responders (*p* = 0.012); a multivariate analysis identified serum IL-6 (OR = 0.72 (95% CI 0.54–0.96), *p* = 0.027) as an independent predictor of IFX therapy response [32]. Another study investigating factors associated with remission of dose escalation in patients with CD showing loss of response to IFX treatment reported that IL-6 ≤ 2.41 pg/mL was one of the factors significantly associated with remission at week 40 [33]. Also, in patients with acute severe UC, IL-6 was able to predict intravenous corticosteroid treatment failure, showing that the risk of failure increased by 40% with each pg/mL increment in IL-6 level [34].

The present study is limited by the size and heterogeneity of the population analyzed, making it difficult to perform any sub-analysis. However, concerning the prediction of therapy response, we corrected for disease entity, type of biologic treatment, and disease activity in order to reduce possible confounding factors. In addition, the prospective design contributed to a reduction in potential biases. We chose, as the primary outcome, steroid-free clinical remission without stopping the biological drug, instead of an endoscopic evaluation, to perform an intention-to-treat analysis that also included patients who did not have a colonoscopy performed after 12 months of therapy due to drug swapping or surgical therapy and because colonoscopy evaluation has a fair number of limitations in CD located in the small bowel (the disease location is not always reachable, or it might be transmural disease). Finally, we have not assessed the concentration of free cytokines in circulation or bound to the corresponding drug and their potential role in the course of biologic treatment. Since this aspect may be clinically relevant, further studies are warranted to address this issue.

## 5. Conclusions

In conclusion, in patients with CD and UC treated with biologics, the reduction of serum IL-6 values from baseline to 10 weeks of treatment may allow for the prediction of clinical response at 12 months of therapy and thus may help clinicians to tailor personalized treatment strategies. Further studies are needed to validate these results in larger groups of patients with IBD undergoing treatment with biologic agents.

## Figures and Tables

**Figure 1 jcm-09-00800-f001:**
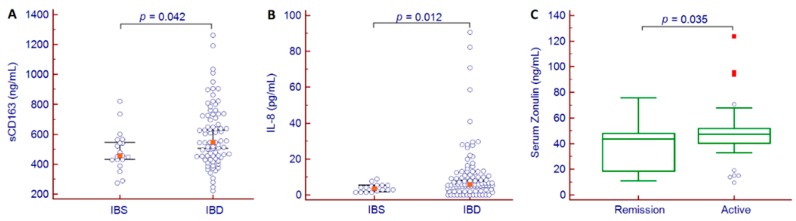
Comparison of sCD163 (**A**) and IL-8 (**B**) values between patients with IBS and those with IBD and serum zonulin (**C**) between IBD patients in remission and those with active disease. Abbreviations: interleukin (IL), irritable bowel syndrome (IBS), inflammatory bowel disease (IBD), soluble CD163 (sCD163).

**Figure 2 jcm-09-00800-f002:**
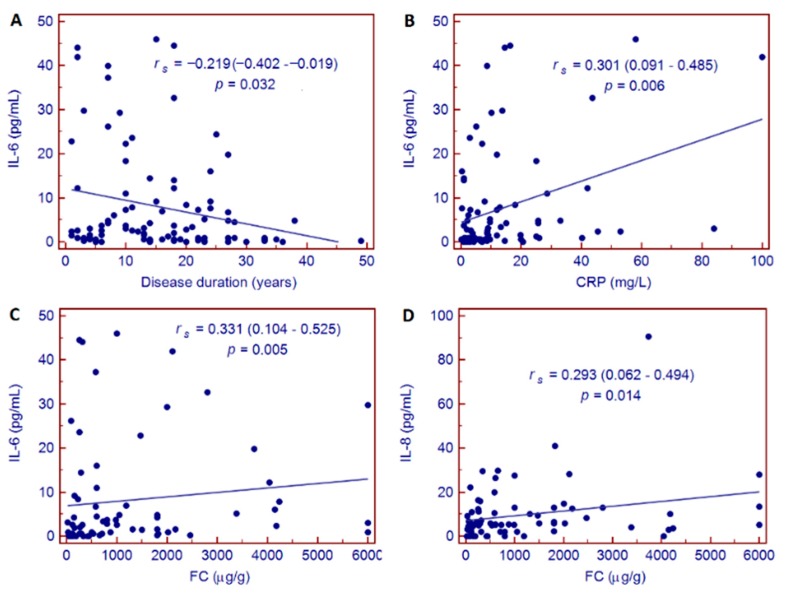
Correlation between IL-6 and disease duration (**A**), CRP values (**B**) and FC concentration (**C**); and between IL-8 and FC (**D**) in patients with IBD. Abbreviations: C reactive protein (CRP), fecal calprotectin (FC).

**Figure 3 jcm-09-00800-f003:**
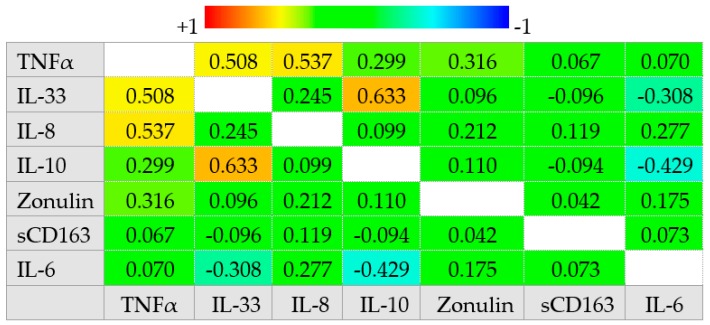
Correlogram reporting the Spearman correlation coefficient between serum zonulin, sCD163, and cytokine values in patients with IBD. Abbreviations: tumor necrosis factor-alpha (TNFα).

**Figure 4 jcm-09-00800-f004:**
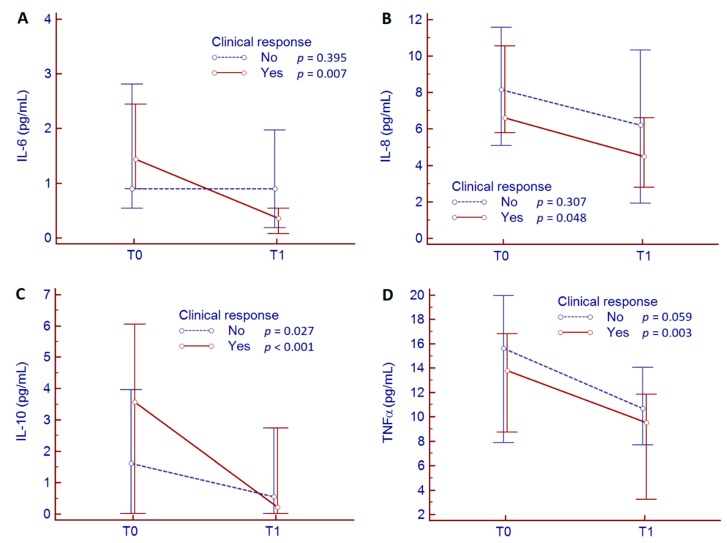
Reduction of IL-6 (**A**), IL-8 (**B**), IL-10 (**C**), and TNFα (**D**) values between baseline (T0) and 10 weeks (T1) of biologic therapy according to the response to treatment.

**Figure 5 jcm-09-00800-f005:**
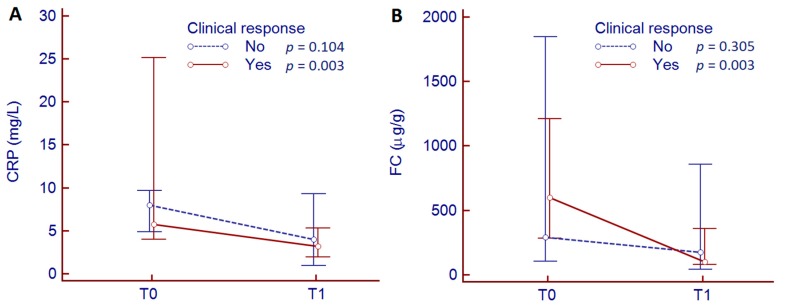
Reduction of CRP (**A**) and FC (**B**) values between baseline (T0) and 10 weeks (T1) of biologic therapy according to treatment response.

**Table 1 jcm-09-00800-t001:** Characteristics of the study population included in the cross-sectional analysis.

	IBD (*n* = 101)	IBS (*n* = 19)	*p*-Value
Age (years), median (range)	48 (18–80)	43 (18–69)	0.067
Gender (M/F)	62/39	6/13	0.023
Disease (CD/UC)	72/29		
Montreal classification			
CD (L1/L2/L3)	24/4/43		
UC (E1/E2/E3)	4/9/16		
Clinical activity			
Remission	27		
Mild	40		
Moderate	28		
Severe	6		
Disease duration (years), median (range)	14 (1–49)		
Previous surgery (yes/no)	48/53		
Smoke (current/never/ex)	23/42/36		
Biochemical activity			
FC (µg/g), median (95% CI)	602 (395–997)		
CRP (mg/L), median (95% CI)	7.0 (5.0–9.0)		
ESR (positive/negative/N/A)	45/35/21		

Abbreviations: inflammatory bowel disease (IBD), irritable bowel syndrome (IBS), male (M), female (F), Crohn’s disease (CD), ulcerative colitis (UC), ileum (L1), colon (L2), ileum + colon (L3), rectum (E1), left side (E2), extensive (E3), fecal calprotectin (FC), confidence interval (CI), C reactive protein (CRP), erythrocyte sedimentation rate (ESR), N/A (not available).

**Table 2 jcm-09-00800-t002:** Comparison of zonulin, sCD163, and cytokine levels at baseline between patients with IBD and those with IBS.

	IBD (*n* = 101)	IBS (*n* = 19)	*p*-Value
Zonulin (ng/mL), median (95% CI)	45.3 (43.5–47.8)	43.3 (37.2–46.4)	0.322
sCD163 (ng/mL), median (95% CI)	547 (506–629)	456 (432–548)	0.042
IL-1β (pg/mL), median (95% CI)	n.q.	n.q.	
IL-4 (pg/mL), median (95% CI)	n.q.	n.q.	
IL-6 (pg/mL), median (95% CI)	2.52 (1.43–4.27)	4.23 (2.87–7.06)	0.140
IL-8 (pg/mL), median (95% CI)	5.79 (5.15–7.96)	3.47 (1.97–5.32)	0.012
IL-10 (pg/mL), median (95% CI)	0.01 (0.01–1.12)	n.q.	
IL-12(p70) (pg/mL), median (95% CI)	n.q.	n.q.	
IL-17 (pg/mL), median (95% CI)	n.q.	n.q.	
IL-23 (pg/mL), median (95% CI)	n.q.	n.q.	
IL-33 (pg/mL), median (95% CI)	0.18 (0.01–15.50)	n.q.	
IFNγ (pg/mL), median (95% CI)	n.q.	n.q.	
TNFα (pg/mL), median (95% CI)	10.48 (8.30–14.21)	5.69 (2.23–11.09)	0.159

Abbreviations: inflammatory bowel disease (IBD), irritable bowel syndrome (IBS), interleukin (IL), interferon-gamma (IFNγ), soluble CD163 (sCD163), tumor necrosis factor-alpha (TNFα), not quantifiable (n.q.), confidence interval (CI).

**Table 3 jcm-09-00800-t003:** Comparison between baseline (T0) and 10 weeks (T1) values of zonulin, sCD163, and cytokines in the 60 IBD patients treated with biologic therapy and according to treatment response.

	*n*	T0	T1	*p*-Value
Zonulin (ng/mL), median (95% CI)	60	46.0 (43.4–49.3)	45.6 (42.0–50.9)	0.722
Responders	32	44.2 (41.8–48.0)	43.1 (39.4–50.2)	0.981
Non-responders	28	48.6 (43.3–53.4)	48.7 (42.1–54.0)	0.568
sCD163 (ng/mL), median (95% CI)	60	520 (464–603)	567 (498–607)	0.818
Responders	32	503 (450–560)	515 (447–599)	0.838
Non-responders	28	552 (465–652)	607 (500–670)	0.864
IL-6 (pg/mL), median (95% CI)	60	1.08 (0.71–2.32)	0.54 (0.18–1.09)	0.013
Responders	32	1.44 (0.90–2.45)	0.36 (0.07–0.54)	0.007
Non-responders	28	0.90 (0.54–2.81)	0.90 (0.18–1.97)	0.395
IL-8 (pg/mL), median (95% CI)	60	7.30 (5.79–10.02)	5.04 (3.19–6.86)	0.043
Responders	32	6.62 (5.79–10.56)	4.50 (2.79–6.62)	0.048
Non-responders	28	8.15 (5.09–11.57)	6.22 (1.92–10.33)	0.307
IL-10 (pg/mL), median (95% CI)	60	1.89 (0.48–4.95)	0.26 (0.01–2.74)	<0.001
Responders	32	3.56 (0.01–6.05)	0.23 (0.01–2.74)	<0.001
Non-responders	28	1.61 (0.01–3.96)	0.56 (0.01–2.74)	0.027
IL-33 (pg/mL), median (95% CI)	60	8.76 (0.09–43.16)	4.64 (0.01–33.56)	0.126
Responders	32	8.76 (0.07–49.86)	4.64 (0.01–46.01)	0.433
Non-responders	28	16.87 (0.01–57.87)	0.01 (0.01–36.71)	0.151
TNFα (pg/mL), median (95% CI)	60	14.46 (9.78–16.76)	10.40 (7.86–11.86)	<0.001
Responders	32	13.79 (8.75–16.82)	9.51 (3.25–11.86)	0.003
Non-responders	28	15.60 (7.86–19.97)	10.69 (7.67–14.04)	0.059

Abbreviations: interleukin (IL), tumor necrosis factor-alpha (TNFα), confidence interval (CI).

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
