# Peer review of "On-Treatment Decrease of Serum Interleukin-6 as a Predictor of Clinical Response to Biologic Therapy in Patients with Inflammatory Bowel Diseases"

_jcm, 2020, doi:10.3390/jcm9030800_

Round 1
Reviewer 1 Report
This is a well written manuscript and would be an important contribution to the literature. I have just a few concerns.
Im not sure the IBS group is really warranted/necessary. When the authors describe the purpose of the study it was to identify biomarkers that may be able to predict therapeutic success of certain biologic drugs in patients with IBD. Im not sure the IBS group really serves as a control for the main purpose of the study as the patients in the IBS group did not have their biomarkers measured over time. Really the IBS patients seem to be there to compare the biomarker concentrations at baseline (and the elevated zonulin in IBS patients is interesting). Please better explain why you included the patients with IBS in this study. Or please state in the introduction that one of the purposes of the study was to compare these markers in patients with IBD vs. IBS?
Line 97- Can you please elaborate on what constituted an indication for biologic treatment? Please define. In other words, how was it determined which patients received biologics?
Line 116-119- These two sentences should be made clearer.
Table 2- specify that these are the concentrations at baseline.
Line 186- please explain what constituted 'active disease'
Line 189-192- how were the drugs the patients placed on determined? physician preference?
Line 193- Could you include date on the percentage of responders to the individual drugs?
Figure 4 & 5- show p-values in the figures?
Line 264- what kind of controls were used in this study (reference 11)?
Author Response
Reviewer #1
This is a well written manuscript and would be an important contribution to the literature. I have just a few concerns.
Im not sure the IBS group is really warranted/necessary. When the authors describe the purpose of the study it was to identify biomarkers that may be able to predict therapeutic success of certain biologic drugs in patients with IBD. Im not sure the IBS group really serves as a control for the main purpose of the study as the patients in the IBS group did not have their biomarkers measured over time. Really the IBS patients seem to be there to compare the biomarker concentrations at baseline (and the elevated zonulin in IBS patients is interesting). Please better explain why you included the patients with IBS in this study. Or please state in the introduction that one of the purposes of the study was to compare these markers in patients with IBD vs. IBS?
According to reviewer suggestion, in the introduction section, we added that the secondary aim of the study was to compare the biomarkers levels between patients with IBD and patients with irritable bowel syndrome (IBS).
Line 97- Can you please elaborate on what constituted an indication for biologic treatment? Please define. In other words, how was it determined which patients received biologics?
Dear reviewer, thank you for your comment. We treated with biologics, according to ECCO guidelines, patients with moderate-to-severe disease activity or steroid-dependent disease with previous failure or intolerance to thiopurines.
Line 116-119- These two sentences should be made clearer.
According to reviewer suggestion, we reworded the two sentences.
Table 2- specify that these are the concentrations at baseline.
Table 2 heading has been modified accordingly.
Line 186- please explain what constituted 'active disease'
Thank you for your question. We considered patients with active disease patients with moderate or severe disease activity or patients currently treated with systemic steroid. This is the reason because we considered, according to literature [21], clinical response to biologic therapy
- a decrease in Harvey-Bradshaw index (HBI) greater than or equal to 3 or in the partial Mayo (pMAYO) score greater than or equal to 2 (or pMAYO ≤ 1 at month 12) in patients with moderate to severe disease activity
- a HBI ≤ 4 at month 12 or pMAYO ≤ 1 at month 12 in absence of corticosteroid therapy (we considered a stop of steroid as a success for patients that at the moment of the start of the biologic were not in moderate or severe disease activity thanks to systemic steroids)
Line 189-192- how were the drugs the patients placed on determined? physician preference?
Thank you for your question. Anti-TNF therapy was the first line choice for all patients, except for the patients with contraindications to these drugs; VDZ or UTK were the second (or third) line choice, except for the patients with contraindications to anti-TNF therapy.
Line 193- Could you include date on the percentage of responders to the individual drugs?
In detail, 19 out of 33 patients (57.6%) responded to ADA, 1 out of 4 (25.0%) responded to IFX, 1 out of 2 (50%) responded to UTK and 11 out of 20 (55.0%) responded to VDZ (see section 3.2 Longitudinal analysis).
Figure 4 & 5- show p-values in the figures?
P values were added to Figure 4 and 5.
Line 264- what kind of controls were used in this study (reference 11)?
The study compared sCD163 levels between patients with IBD (58 patients with CD, 40 patients with UC) vs and 90 healthy controls HC. The category of control group has been added in the discussion.
Reviewer 2 Report
In the current study, the authors investigated whether serum zonulin, sCD163, and a panel of serum cytokines can be used to predict and monitor response to biologic therapies in patients with Crohn’s disease. They found that a reduction of serum interleukin-6 from week 0 to week 10 was associated with a clinical response at 12 months of therapy.
The authors have to be commended for addressing a clinically relevant question. Although there are some inherent limitations to the study design, these are described clearly in the discussion. Nevertheless, I would like to take this opportunity to clarify a number of other things in more detail:
Major comments:
- It is important to discuss in more detail the choice for patients with irritable bowel syndrome as a control group (instead of healthy controls).
- The authors mention that patients who were lost to follow-up were considered as treatment failures (i.e., single imputation for handling missing data). How many patients were lost to follow-up? A sensitivity analysis may be appropriate.
- Sixty patients with active disease underwent biologic therapy (cf. Results: 3.2. Longitudinal analysis). Patients affected by IBD with an indication to biologic treatment were further included in the longitudinal analysis (cf. Materials and Methods: 2.1. Patients). From Table 1, it can be seen that 34 (not 60) patients have moderate-to-severe IBD and thus qualify for biological therapy. Please clarify this discrepancy.
- In the multivariate logistic regression analysis, please also correct for baseline disease activity.
Minor comments:
- A total of 43 patients had a diagnosis of CD (cf. Results: 3.2. Longitudinal analysis). Among patients with CD, 31 underwent treatment with adalimumab, 10 with vedolizumab, and 1 with ustekinumab = 42 Please clarify.
- Does the TNFα immunoassay quantify only free molecules or also those bound to infliximab/adalimumab? Assay characteristics impact the measurement results. Please address them in detail in the discussion of the manuscript to allow a better understanding of the findings, and for comparability with other studies using different assays.
Author Response
Reviewer #2
In the current study, the authors investigated whether serum zonulin, sCD163, and a panel of serum cytokines can be used to predict and monitor response to biologic therapies in patients with Crohn’s disease. They found that a reduction of serum interleukin-6 from week 0 to week 10 was associated with a clinical response at 12 months of therapy.
The authors have to be commended for addressing a clinically relevant question. Although there are some inherent limitations to the study design, these are described clearly in the discussion. Nevertheless, I would like to take this opportunity to clarify a number of other things in more detail:
Major comments:
It is important to discuss in more detail the choice for patients with irritable bowel syndrome as a control group (instead of healthy controls).
IBD and IBS are two conditions profoundly different from a pathophysiological and prognostic point of view but both conditions may share overlapping clinical features such as abdominal pain, excessive flatus, bloating and altered bowel habit. Indeed, up to 40% of patients with IBD have the same presentation as patients with IBS (Keohane J, et al. Am J Gastroenterol 2010). Since in clinical practice a considerable proportion of patients with IBS are referred to secondary/tertiary care services, in our opinion, patients with IBS represent a more suitable control group in comparison to healthy subjects for studies involving patients with IBD.
The authors mention that patients who were lost to follow-up were considered as treatment failures (i.e., single imputation for handling missing data). How many patients were lost to follow-up? A sensitivity analysis may be appropriate.
A total of 28 patients did not respond to treatment. Switching to other biologic, addition of a corticosteroid and surgical resection were considered criteria of treatment failure. Only 2 patients (3.3%) were lost to-follow up (LTFU) and were included in the intention to treat (ITT) analysis. By per Protocol (pP) analysis, we observed that IL-6 reduction from baseline to 10 weeks of treatment was still an independent predictor of clinical response at 12 months of therapy (OR=5.08, 95%CI 1.33-19.41, p=0.017). Due to the low number of patients LTFU and the consistent results between ITT and pP analysis, we did not add this result in the manuscript.
Sixty patients with active disease underwent biologic therapy (cf. Results: 3.2. Longitudinal analysis). Patients affected by IBD with an indication to biologic treatment were further included in the longitudinal analysis (cf. Materials and Methods: 2.1. Patients). From Table 1, it can be seen that 34 (not 60) patients have moderate-to-severe IBD and thus qualify for biological therapy. Please clarify this discrepancy.
Dear reviewer, thank you for your question. We treated with biologics, according to ECCO guidelines, patients with moderate-to-severe disease activity or steroid-dependent disease with previous failure or intolerance to thiopurines. We considered patients with active disease patients with moderate or severe disease activity or patients currently treated with systemic steroid. This is the reason because we considered, according to literature [21], clinical response to biologic therapy
- a decrease in Harvey-Bradshaw index (HBI) greater than or equal to 3 or in the partial Mayo (pMAYO) score greater than or equal to 2 (or pMAYO ≤ 1 at month 12) in patients with moderate to severe disease activity
- a HBI ≤ 4 at month 12 or pMAYO ≤ 1 at month 12 in absence of corticosteroid therapy (we considered a stop of steroid as a success for patients that at the moment of the start of the biologic were not in moderate or severe disease activity thanks to systemic steroids)
In the multivariate logistic regression analysis, please also correct for baseline disease activity.
According to reviewer suggestion, we performed a multivariate logistic regression analysis including also disease activity among independent variables. IL-6 reduction from baseline to 10 weeks of treatment remained an independent predictor of clinical response at 12 months of therapy (OR=4.75, 95%CI 1.25-18.02, p=0.022) (see lines 245-248).
Minor comments:
A total of 43 patients had a diagnosis of CD (cf. Results: 3.2. Longitudinal analysis). Among patients with CD, 31 underwent treatment with adalimumab, 10 with vedolizumab, and 1 with ustekinumab = 42 Please clarify.
The number of patients treated with the corresponding biologic agent has been corrected (see lines 198-199).
Does the TNFα immunoassay quantify only free molecules or also those bound to infliximab/adalimumab? Assay characteristics impact the measurement results. Please address them in detail in the discussion of the manuscript to allow a better understanding of the findings, and for comparability with other studies using different assays.
To the best of our knowledge, the TNFα immunoassay used in our study detects total TNFα (free and bound to IFX/ADA). Several studies showed increased TNFα concentrations shortly after initiation of anti-TNF treatment followed by a gradual on-treatment decline. This may be explained by the fact that TNF bound to a TNF inhibitor has a prolonged half-life, because the TNF-inhibiting antibodies themselves have a very long half-life of several weeks. In our study we did not evaluated early on-treatment cytokines concentrations therefore we did not notice this phenomenon. Taking into account these considerations, we added into the study limitations paragraph of the discussion section, that we did not investigate the concentrations of free and bound cytokines to the corresponding drug. Further study may investigate this issue.